# Synthesis of NiW Supported on an Al-Modified Cubic Ia3d Mesoporous KIT-5 Catalyst and Its Hydrodenitrogenation Performance of Quinoline

**Xing Liu [1,2], Shaoqing Guo [3], Xin Li [1], Lijing Yuan [1], Hongyu Dong [1], Zhenrong Li [1], Haitao Cui [1,* and Liangfu Zhao [1,*]**

[1] Institute of Coal Chemistry, Chinese Academy of Sciences, Taiyuan 030001, China; liuxing@sxicc.ac.cn (X.L.); lixin@sxicc.ac.cn (X.L.); yuanlijing@sxicc.ac.cn (L.Y.); donghongyu@sxicc.ac.cn (H.D.); lizr@sxicc.ac.cn (Z.L.)

[2] University of the Chinese Academy of Sciences, Beijing 100039, China

[3] College of Environment and Safety, Taiyuan University of Science and Technology, Taiyuan 030024, China; guosq@tyust.edu.cn

* Correspondence: cuiht@sxicc.ac.cn (H.C.); lfzhao@sxicc.ac.cn (L.Z.)

**Abstract:** Pure KIT-5 and a series of Al-KT-X materials modified by different amounts of aluminum were synthesized by a direct hydrothermal method and acted as supports for the catalysts of a quinoline hydrodenitrification reaction with the NiW active phases supported. The results of X-ray diffraction (XRD), $N_2$ isotherm absorption-desorption, scanning electron microscopy (SEM), and Fourier transform infrared (FTIR) for the supports indicated that Al species were embedded into the framework of the KIT-5 materials with a large pore size, pore volume, and specific surface area. The Pyridine-Fourier transform infrared spectroscopy (Py-IR) result of the catalysts demonstrated that the addition of aluminum atoms enhanced the acidity of the catalysts. The results of the high-resolution transmission electron microscopy (HRTEM) and X-ray photoelectron spectra (XPS) characterizations for the sulfide catalysts indicated that the embedded Al species could facilitate the dispersion of active metals and the formation of the active phases. Among all the catalysts, NiW/Al-KT-40 showed the maximal hydrodenitrogenation conversion ($HDN_C$) due to its open three-dimensional pore structure, appropriate acidity, and good dispersion of active metals.

**Keywords:** hydrodenitrogenation; quinoline; KIT-5 mesoporous material; Al-modified support

## 1. Introduction

With the world's light oil reserves decreasing, the conversion of unconventional oil products, such as coal tar and heavy oil, into clean fuel oil has received increasing attention from researchers. The hydrofining of coal tar to produce light fuel oil is of great practical and strategic significance to replace certain petroleum resources. However, coal tar is rich in polycyclic aromatic hydrocarbons, colloids, asphalts, and a large number of impurity elements, such as metals, sulfur, and nitrogen, which can cause a negative effect on the further utilization of coal tar [1–3]. In particular, the presence of nitrogen compounds in car tar not only produces NOx pollutants during the combustion process, but also deactivates the catalysts of the hydrocracking or hydrorefining process. Hench, nitrogen compounds in car tar must be removed by hydrodenitrogenation (HDN) reaction, and developing efficient hydrogenation catalysts is one of the key technologies for the HDN of coal tar [4–9].

Currently, scientists have made many attempts to prepare high performance catalysts, including the application of different active phases and various supports. The traditional hydrogenation catalysts supported molybdenum sulfide or tungsten sulfide with a Ni (Co) atom as the promoter are extensively applied in industry. NiW, as the active phase of catalysts, has excellent catalytic activity on the

HDN performances, especially under harsh reaction conditions with higher hydrogen pressure and temperature [10–15]. Generally, $\gamma$-$Al_2O_3$ has been widely used as a conventional support of HDN catalysts due to its excellent mechanical performances, low price, and high thermal and hydrothermal stability [3,16]. However, the small surface area and single Lewis acid site distribution restrict the hydrogenation activity [17–19]. Thus, modifications of $\gamma$-$Al_2O_3$ with fluorine or phosphorus have been conducted by researchers [3,10,20]. For example, Shi et al. modified $\gamma$-$Al_2O_3$ with phosphorus and demonstrated that the addition of phosphorus could alter the acid site distributions and improve the HDN catalytic performance to an extent. However, the highest HDN conversion was only 74.36%, which cannot meet the need of commercial application [20]. Guo et al. modified $\gamma$-$Al_2O_3$ with fluorine and found that the addition of fluoride decreased the specific surface area of the catalyst, which cannot significantly improve the HDN activity [3].

To further improve the catalytic performance, the development of advanced supports for the HDN catalysts became a focus of scientific research. Certain mesoporous materials, such as MCM-48, SBA-15, and FDU-12, with orderly mesoporous structures, large specific surface areas, and uniform pore sizes, became a research hotspot of the catalyst supports [21–27]. In particular, they showed potential for HDN catalytic reactions. However, weak acids limit the HDN activity to an extent.

Studies have shown that heteroatom introduction, including $Zr^{4+}$, $Ti^{4+}$, and $Al^{3+}$, can improve the acidity properties of a catalyst, which will promote the dispersion of active metals [4,28–34]. For example, Shao et al. prepared Al-modified MCM-48 supports for the NiW catalyst in the hydrodenitrogenation (HDN) reaction of quinoline and the activity results showed that Al-modified NiW/MCM-48 catalysts displayed higher HDN activity than aluminum free NiW/MCM-48, since the introduction of suitable aluminum atoms enhanced the acidity of the support and, hence, improved the sulfidation degree of the catalyst [27].

As a novel mesoporous molecular sieve, KIT-5 can be a candidate for the support of HDN catalysts because it possesses an excellent face-centered-cubic *Fm3m* symmetry structure with a large specific surface area and adjustable pore diameter [22–24]. However, its weak acidity does not favor the HDN reaction. Thus, the aluminum atoms should be introduced to the KIT-5 material to improve its acidity and HDN activity [35–38].

In this study, a series of Al-modified KIT-5 supports with different silicon-aluminum ratios (10, 40, 80, and 200) as well as the pure KIT-5 supports were successfully fabricated by the one-step direct hydrothermal method. In addition, the corresponding NiW/Al-KT-X catalysts were synthesized with the incipient impregnation method. The catalysts were characterized by X-ray diffraction (XRD), $N_2$ absorption−desorption, scanning electron microscopy (SEM), transmission electron microscopy (TEM), Pyridine-Fourier transform infrared spectroscopy (Py-IR), and X-ray photoelectron spectra (XPS). The HDN activity of the catalysts was evaluated in a fixed bed with quinoline as the reactant under different reaction conditions.

## 2. Results

### 2.1. Structural Characteristics of the Supports

### 2.1.1. Small-Angle XRD Characterization

Figure 1 shows the XRD patterns of aluminum free KIT-5 and the Al-KT-X samples (X represents Si/Al ratios of 200, 80, 40, and 10). Clearly, the XRD pattern for the aluminum free KIT-5 sample in Figure 1 has two strong diffraction peaks located at about 0.7° and 0.8°, corresponding to the (211) and (220) lattice planes indexed to the highly ordered three-dimensional face-centered cubic *Fm3m* symmetry structure [35]. The modified Al-KT-X samples also exhibited similar diffraction peaks, indicating that they possess a similar ordering mesoporous structure after the introduction of aluminum atoms. However, compared with the pure KIT-5 support, the intensity of the two diffraction peaks of the Al-modified supports clearly declined, confirming that the introduction of aluminum atoms reduced the ordering of the mesoporous molecular sieves to some extent.

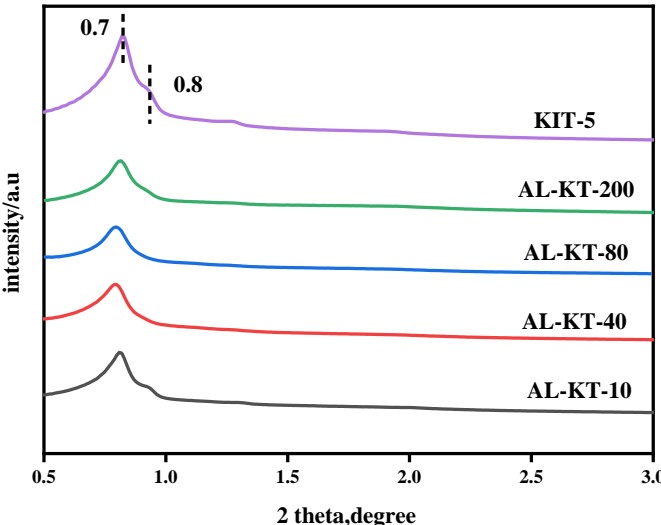

**Figure 1.** Small-angle X-ray diffraction (XRD) patterns of Al-KT-X and KIT-5 materials.

### 2.1.2. $N_2$ Adsorption–Desorption Characterization

The $N_2$ adsorption–desorption isotherms of the KIT-5 and Al-KT-X samples are presented in Figure 2A. The $N_2$ isotherms of all the samples possess the type IV hysteresis loops at relative pressure ranging from 0.40–0.70, which corresponds to the typical cage-type mesoporous structure [39]. From Figure 2A, the hysteresis loops of all samples are basically similar, demonstrating that the ordered mesoporous structures of Al-modified materials are still reserved after the introduction of aluminum atoms. These results are consistent with the small-angle XRD results. The pore size distribution (PSD) curves of the samples are shown in Figure 2B. Compared with the pure KIT-5 sample, each Al-modified support possessed a larger pore diameter.

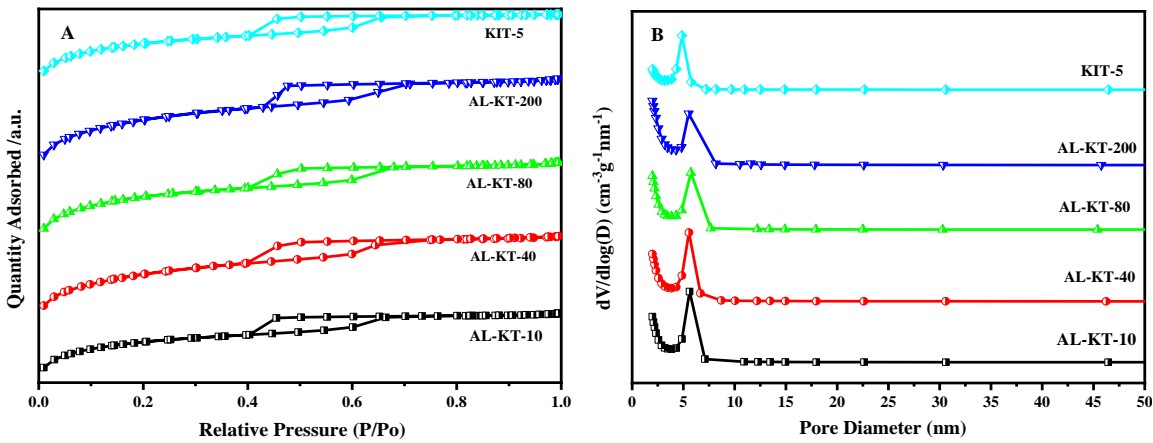

**Figure 2.** (**A**) The $N_2$ adsorption-desorption isotherms and (**B**) pore size distributions of the supports.

The detailed textural properties of the as-synthesized supports are summarized in Table 1. Compared with the Al-modified supports, the pure KIT-5 support exhibited the smallest specific surface area (790.5 m$^2$ g$^{-1}$), pore volume (0.36 cm$^3$ g$^{-1}$), and pore diameter (4.75 nm), because the cation radius of Al$^{3+}$ (0.053 nm) is larger than that of Si$^{4+}$ (0.040 nm), resulting in the expansion of crystals of Al-modified supports [26,40,41]. A similar phenomenon was observed by Cao, who found that the pore size, specific surface, and pore volume for the Ti-modified supports became larger than the pure support [23]. The specific surface and pore volume for the modified Al-KT-X supports decreased with increasing aluminum content. This can be explained by the mesoporous channel being plugged due to the introduction of more aluminum atoms [27].

**Table 1.** The pore structural properties of supports.

| Samples | $S_{BET}$ (m$^2$ g$^{-1}$) | V (cm$^3$ g$^{-1}$) | D (nm) |
|---------|------------------------------|----------------------|--------|
| AL-KT-10 | 1012.89 | 0.62 | 5.51 |
| AL-KT-40 | 1057.15 | 0.64 | 5.43 |
| AL-KT-80 | 1087.57 | 0.67 | 5.66 |
| AL-KT-200 | 1121 | 0.72 | 5.43 |
| KIT-5 | 790.5 | 0.36 | 4.75 |

### 2.1.3. Fourier Transform Infrared (FTIR) Characterization of Materials

FTIR analysis was performed to further verify the introduction of aluminum to KIT-5 molecular sieves. As shown in Figure 3, all the synthesized materials possessed the characteristic vibration peaks of the silicon-oxygen tetrahedron skeleton structure at about 460, 806, and 1080 cm$^{-1}$ [42,43]. The pure KIT-5 support exhibited the weak vibration mode at 950 cm$^{-1}$ due to the Si–OH stretching vibration [40,44]. However, for the modified Al-KT-X materials, the peak at 950 cm$^{-1}$ was stronger than that for pure KIT-5 due to the synergistic effect of the Si–OH and Si–O–Al stretching vibrations. The intensity of this peak at 950 cm$^{-1}$ increased progressively with the increasing aluminum content, indicating that the aluminum atoms were successfully incorporated into the KIT-5 framework [40]. Compared with the pure KIT-5 samples, the Al-modified supports showed a slight blue shift for all peaks, indicating that the aluminum atoms were embedded into the framework of the KIT-5 support [27].

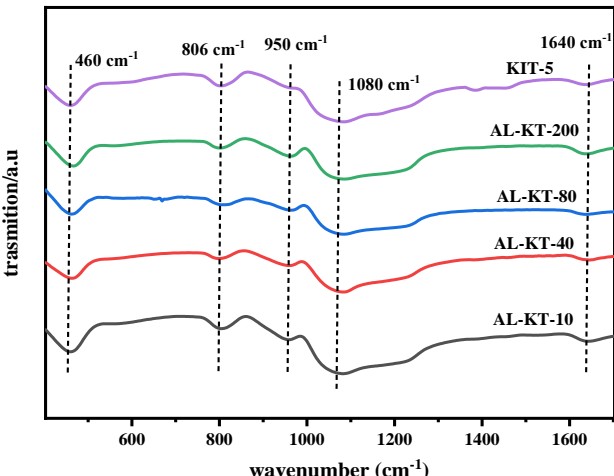

**Figure 3.** Fourier transform infrared (FTIR) spectra of the KIT-5 support and the Al-KT-X supports.

### 2.1.4. SEM and Energy Dispersive Spectroscopy (EDS) of the Supports

Figure 4 shows the scanning electron microscopy (SEM) morphologies of the samples. The morphologies of all the supports are clearly irregular. The surface of the pure KIT-5 is smoother than that of the Al-modified materials, because the surface of the Al-modified samples is damaged to some extent with the introduction of the aluminum atoms [45]. The Al-KT-40 sample was also analyzed by energy dispersive spectroscopy (EDS), and the result is presented in Figure 5. The EDS patterns show three peaks for Si, Al, and O, implying the introduction of aluminum atoms into the KIT-5 support. In addition, we observed that aluminum was uniformly distributed on the surface of the Al-KT-40 material.

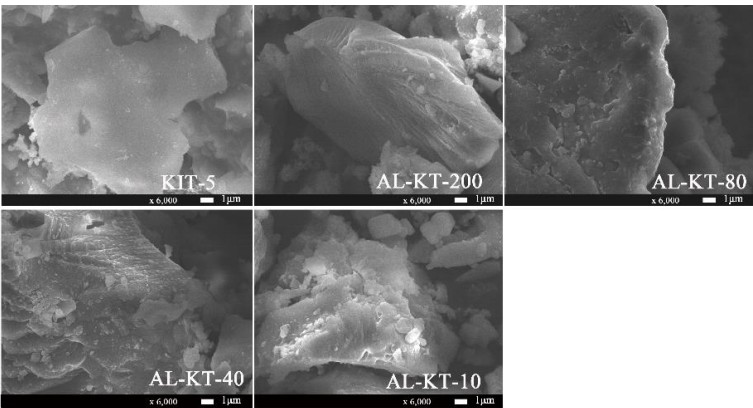

**Figure 4.** Scanning electron microscopy (SEM) images of the Al-KT-X and KIT-5 supports.

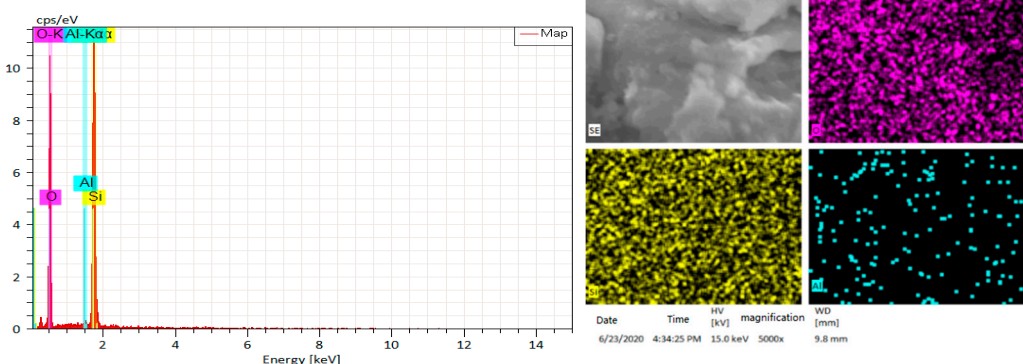

**Figure 5.** The energy dispersive spectroscopy (EDS) pattern of the Al-KT-40 sample-corresponding electron mapping.

### 2.1.5. TEM of the Supports

Transmission electron microscopy (TEM) images for all samples are shown in Figure 6 to observe the ordered channels of mesoporous materials. From Figure 6, it can be clearly seen that the Al-modified supports show similar highly long-range ordering channel structures with the pure KIT-5 support. The interplanar spacing with different supports was measured using the Nano Measurer and a similar result (~6 nm) was obtained. This indicates that the structure of the Al-modified materials was not destroyed by the introduction of aluminum atoms [35,45]. These results agree well with the $N_2$ adsorption and XRD results.

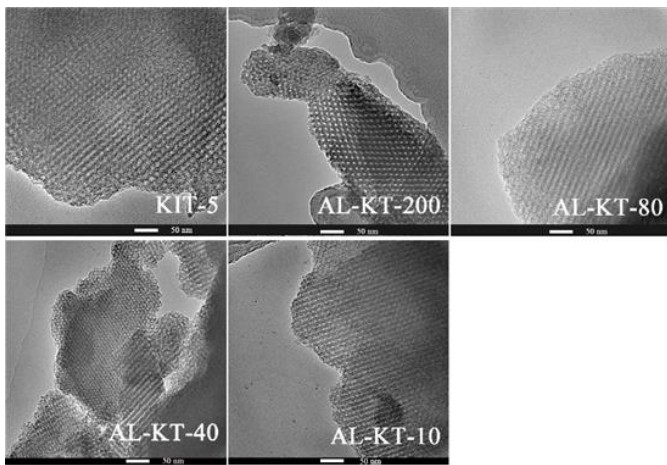

**Figure 6.** The transmission electron microscopy images of the Al-KT-X and KIT-5 supports.

## 2.2. Characteristics of the Catalysts

### 2.2.1. Wide-Angle XRD of the Catalysts

The wide-angle XRD patterns for the NiW/Al-KT-X and NiW/KIT-5 catalysts are shown in Figure 7. The diffraction pattern of the NiW/KIT-5 catalyst has clear characteristic diffraction peaks corresponding to $WO_3$ (PDF No. 71-0131), $NiWO_4$ (PDF No. 72–1189) and NiO (PDF No. 89–8397), indicating the formation of larger crystallites. However, with an increase in the aluminum content in the samples, all peak intensities over Al-modified catalysts significantly decreased, indicating that the active species had smaller crystallites and better dispersion. This is due to a stronger interaction between the supports and active phases and avoiding the formation of agglomerates [24].

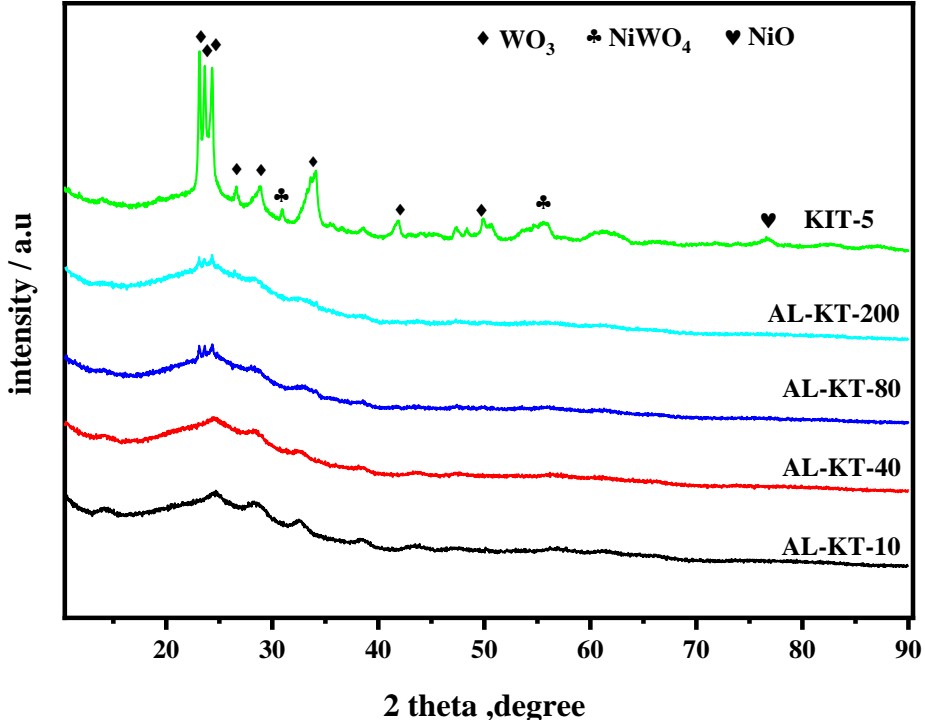

**Figure 7.** Wide-angle XRD patterns of the NiW/Al-KT-X and NiW/KIT-5 catalysts.

### 2.2.2. Py-IR of Various Catalysts

Py-IR was performed to analyze the acidity of catalysts quantitatively and qualitatively, and the results are shown in Figure 8. The band at approximately 1490 cm$^{-1}$ is attributed to Brønsted and Lewis acids, the bands at about 1450 cm$^{-1}$ and 1608 cm$^{-1}$ are assigned to Lewis acid sites, and the bands at both 1639 cm$^{-1}$ and 1540 cm$^{-1}$ result from Brønsted acid sites [8,24,27]. The number of pyridine molecules desorbed at 100, 200, and 300 °C stands for the amounts of weak, intermediate, and strong acid over different catalysts, respectively.

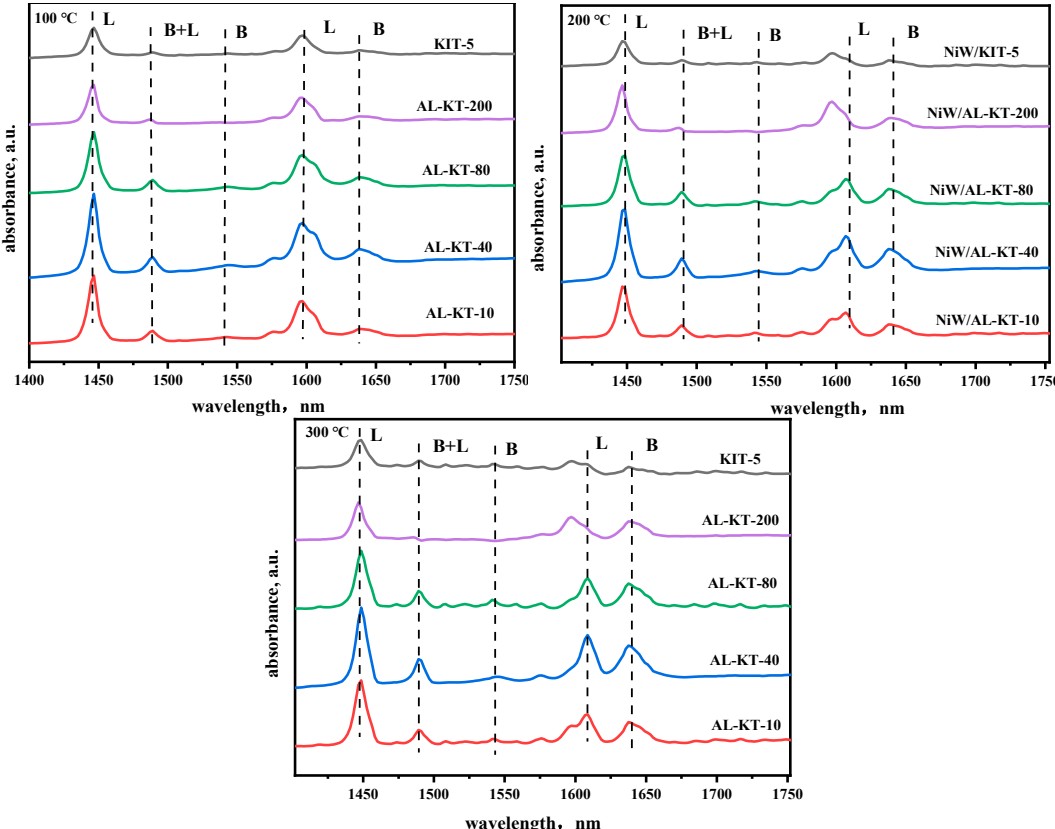

**Figure 8.** Pyridine-Fourier transform infrared spectroscopy (Py-IR) spectra of various catalysts at different temperatures.

The detailed acid strength distribution and calculated acid content of the catalysts are listed in Table 2. We observed that the amount of the Brønsted and Lewis acid sites on the catalysts with Al-modified supports was higher than that on the catalyst with pure support, which agrees with the literature regarding the catalysts with Al, Zr, and Ti-modified supports [24]. The amount of the Brønsted and Lewis acid sites on the catalysts at different temperatures followed the order: NiW/Al-KT-40 > NiW/Al-KT-80 > NiW/Al-KT-10 > NiW/Al-KT-200 > NiW/KIT-5. The minimal amount of Brønsted and Lewis acid sites for NiW/KIT-5 derives from the electrical neutrality of the silicon skeleton structure [23,46]. The higher amount of the Brønsted and Lewis acid sites for the Al-modified catalysts results from the presence of aluminum species, which changes the electron density near the silicon atoms [47]. For the NiW/Al-KT-40 catalyst with the highest amount of the Brønsted and Lewis acid sites, the Lewis acid sites were approximately 2.1–2.6 times as much as that of the NiW/KIT-5 catalyst on different acid intensities, while the amount of Brønsted acid sites was 4.8–8.6 times. The higher increment of Brønsted acid sites was possibly attributed to the different electronegativities between silicon and aluminum atoms, implying that the presence of aluminum species weakens the Si-OH bond and produces the Brønsted acid sites [48].

**Table 2.** Acidity amounts over NiW/Al-KT-X and NiW/KIT-5 catalysts.

| Sample | Brønsted Acidity (µmol/g) | | | Lewis Acidity (µmol/g) | | |
|---|---|---|---|---|---|---|
| | 100 °C | 200 °C | 300 °C | 100 °C | 200 °C | 300 °C |
| NiW/KIT-5 | 2.75 | 2.37 | 2.14 | 125.1 | 61.03 | 39.23 |
| NiW/Al-KT-200 | 4.73 | 2.79 | 2.17 | 128.77 | 90.72 | 39.80 |
| NiW/Al-KT-80 | 23.64 | 15.03 | 10.31 | 320.75 | 149.46 | 83.23 |
| NiW/Al-KT-40 | 24.50 | 15.37 | 10.55 | 327.14 | 152.44 | 84.89 |
| NiW/Al-KT-10 | 12.23 | 6.65 | 3.62 | 239.71 | 109.51 | 75.44 |

### 2.2.3. TEM of Sulfide Catalysts

To observe the morphology of the sulfide catalysts and dispersion of activated phases, high-resolution transmission electron microscopy (HRTEM) was conducted, and the representative micrographs for each sulfide catalyst are shown in Figure 9. As can be seen in the picture, the black filamentary fringes are $WS_2$ slabs approximately 0.64 nanometers apart, which corresponds to the crystalline 002 plane of $WS_2$ species [27]. The staked $WS_2$ layers of the NiW/KIT-5 catalyst are long and bent (sometimes entwined) on the external surface compared with the sulfide catalysts of the Al-modified support, which ascribes to the pore mouths clogging for the NiW/KIT-5 [48]. This is consistent with literature about Zr-modified supports [21]. We found that the NiW/KIT-5 catalyst possessed highly stacked $WS_2$ slabs compared with the Al-modified catalysts, which is attributed to the weaker metal–support interaction [49].

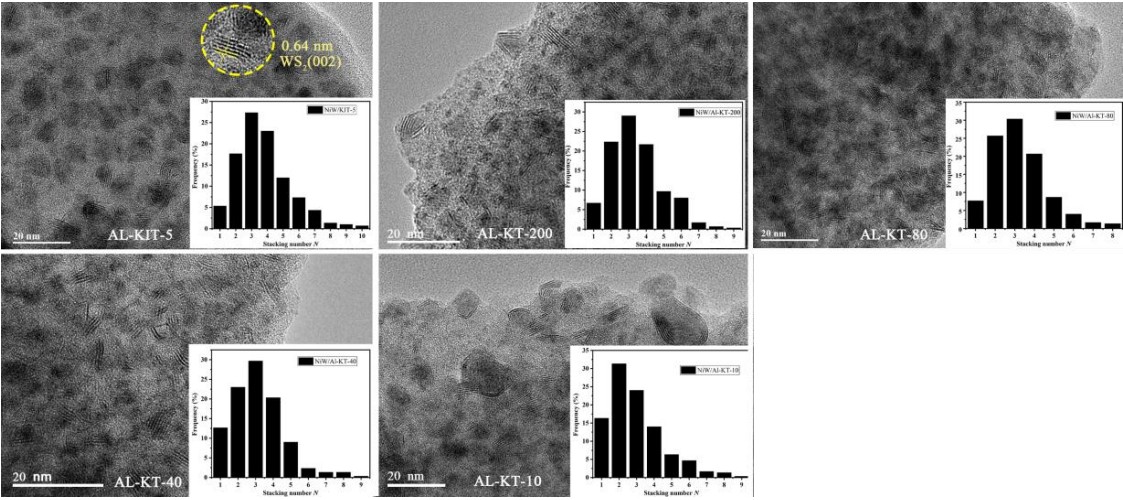

**Figure 9.** High-resolution transmission electron microscopy (HRTEM) micrographs of the sulfide catalysts.

For further analysis, close to 300 stacking layers deriving from 20 photographs for each sulfide catalyst were counted to obtain the average length ($\overline{L}$) and the number ($\overline{N}$) of $WS_2$ stacking layers through statistical analyses, and the corresponding dates are listed in Table 3. The average length and stacking layer number of $WS_2$ crystallites on the different catalysts changed with the change of the support of the catalysts. The average numbers of $WS_2$ slabs on different sulfide catalysts changed in the following sequence: NiW/Al-KT-10 (2.92 nm) < NiW/ Al-KT-40 (3.11 nm) < NiW/Al-KT-80 (3.23 nm) < NiW/Al-KT-200 (3.34 nm) < NiW/KIT-5 (3.75 nm). The minimum stacking numbers for the NiW/Al-KT-10 catalyst was ascribed to the strong interaction between the active phase and the support compared with the other catalysts [8,21,50]. The average length of the $WS_2$ slabs over the sulfide catalysts increased in the order of: NiW/Al-KT-40 (3.60 nm) < NiW/ Al-KT-10 (3.68 nm) < NiW/Al-KT-80 (3.77 nm) < NiW/Al-KT-200 (4.30 nm) < NiW/KIT-5 (5.49 nm). Compared with the NiW/KIT-5 catalyst, all Al-modified sulfide catalysts possessed a shorter length and fewer numbers of $WS_2$ stacking layers, which is attributed to the enhanced interaction between the support and the active metal due to the introduction of aluminum atoms [11].

**Table 3.** HRTEM characterization of sulfide catalysts.

| Catalysts | NiW/KIT-5 | NiW/Al-KT-200 | NiW/Al-KT-80 | NiW/Al-KT-40 | NiW/Al-KT-10 |
|---|---|---|---|---|---|
| $\overline{L}$ | 5.49 | 4.30 | 3.77 | 3.60 | 3.68 |
| $\overline{N}$ | 3.75 | 3.34 | 3.23 | 3.11 | 2.92 |

### 2.2.4. XPS of Sulfide Catalysts

All the sulfide catalysts were characterized with the X-ray photoelectron spectra (XPS) to analyze the chemical state changes and sulfidation degree of W species. The decomposition of W 4f spectra of the series of sulfided catalysts were analyzed using XPS-PEAK software. The results of XPS over the sulfide catalysts are given in Figure 10, in which the W 4f spectra consist of three well-resolved contributions including $W^{4+}$, $W^{5+}$, and $W^{6+}$. The two peaks present at approximately 32.1 and 34.3 eV, with a fixed intensity ratio of 4:3, correspond to W 4f $_{7/2}$ and W 4f $_{5/2}$ in the $W_{4+}$ state species of the $WS_2$ phase, respectively [27]. The peak appearing at 38.0 eV corresponds to the $W^{4+}$ 5p contribution [51]. The vibration peaks with binding energy at approximately 33.2 and 35.2 eV is associated with the $W^{5+}$ species of the $WS_xO_y$ phase [11]. Additionally, the peaks around 35.9 and 37.9 eV are ascribed, respectively, to the W 4f $_{7/2}$ and W 4f $_{5/2}$, which can be assigned to $W^{6+}$ oxide species of $WO_3$ or $NiWO_4$ phase, indicating the part of W species that still exists in the oxidation state after sulfuration [14].

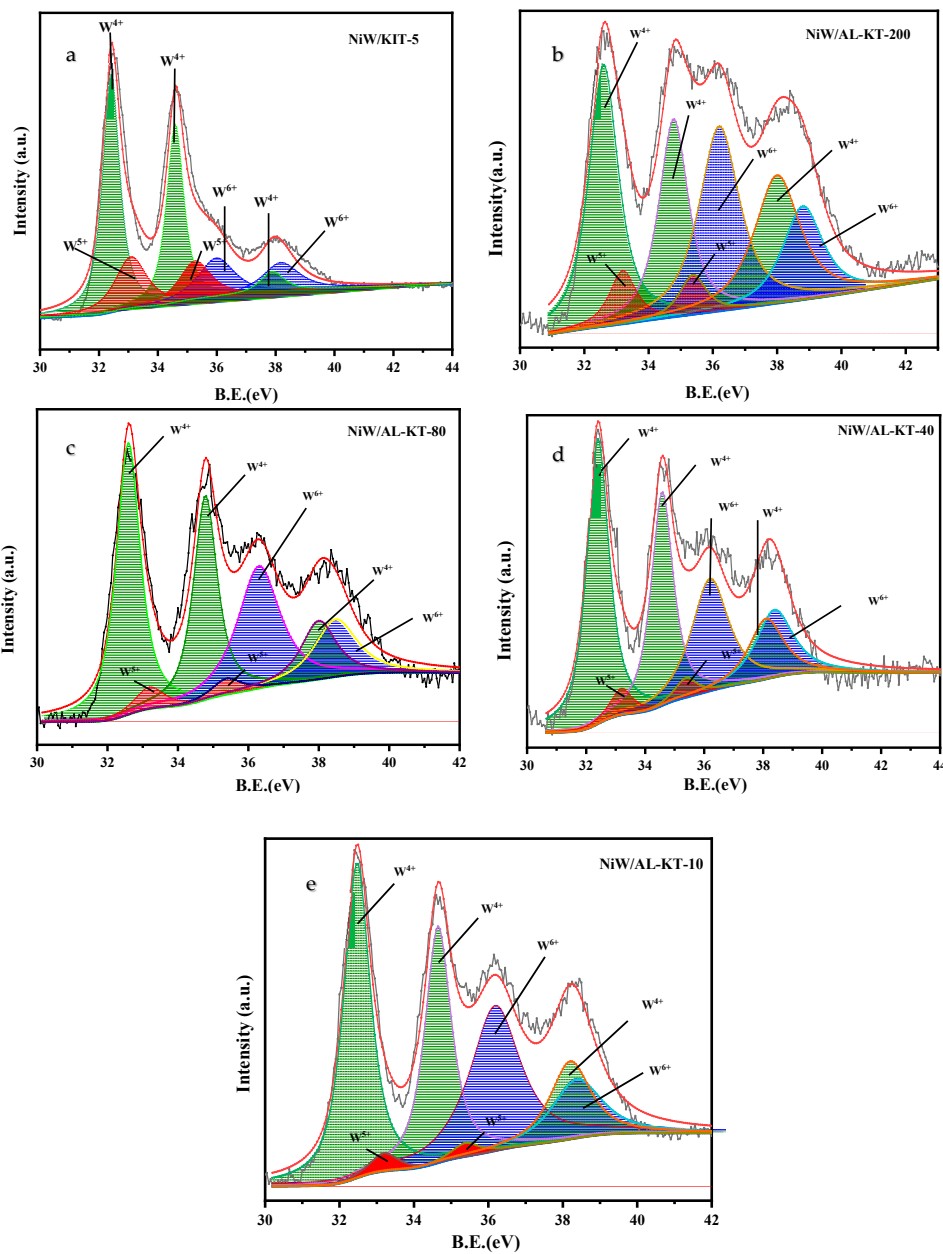

**Figure 10.** W 4f decomposition spectra of NiWS catalysts. (**a**) NiW/KIT-5; (**b**) NiW/KIT-200; (**c**) NiW/KIT-80; (**d**) NiW/KIT-40; (**e**) NiW/KIT-10.

The XPS fitting results of different W species over the sulfide catalysts obtained according to the deconvolution method are listed in Table 4, and the sulfidation degree of W species was determined by the ratio of $W^{4+}$ species to all the W species. The results show that the modified NiW/Al-KT-X catalysts had a higher sulfidation degree than the NiW/KIT-5 catalyst because the introduction of aluminum species regulates the interaction between supports and active metals and, thus, enhances the dispersion of active metals [16]. With the aluminum species increasing, the sulfuration degree of the catalysts enhanced first and then decreased in the order of: NiW/Al-KT-40 (63.16%) > NiW/ Al-KT-80 (61.97%) > NiW/ Al-KT-10 (61.79%) > NiW-Al-KT-200 (58.84%) > NiW/KIT-5 (57.95%). Among all the catalysts, the NiW/Al-KT-40 catalyst exhibited the highest sulfidation degree due to its high acidity and the appropriate texture property [16,52].

**Table 4.** Respective proportions of the W species obtained from decomposition of W 4f spectra of NiWS sulfide catalysts.

| Catalysts | $W^{4+}$ (ar.%) | $W^{5+}$ (ar.%) | $W^{6+}$ (ar.%) |
|---|---|---|---|
| NiW/KIT-5 | 57.95% | 19.25% | 22.80% |
| NiW/Al-KT-10 | 61.79% | 3.11% | 35.10% |
| NiW/AL-KT-40 | 63.16% | 4.34% | 31.95% |
| NiW/AL-KT-80 | 61.97% | 61.97% | 61.97% |
| NiW/AL-KT-200 | 58.84% | 7.54% | 33.99% |

ar.% means the area percentage of the X-ray photoelectron spectra (XPS) peak.

### 2.3. Investigations of Quinoline HDN over Various Catalysts

The results of the catalytic activity of each sulfide catalyst at different temperatures are displayed in Table 5 and Figure 11. The $r_{HDN}$ and k, calculated by formulas 4 and 5, are also listed in Table 5. Clearly, the hydrodenitrogenation conversion (HDNC), k, and $r_{HDN}$ over all catalysts gradually increased with the increasing reaction temperature, indicating that the further reaction of hydrogenation or the breakage of the C−N bond was accelerated with increasing reaction temperature [19,53–56]. The $HDN_C$, $r_{HDN,}$ and k of all the catalysts had the following order: NiW/Al-KT-40 > NiW/Al-KT-80 > NiW/Al-KT-10 > NiW/Al-KT-200 > NiW/KIT-5, illustrating that the introduction of a suitable aluminum specie into KIT-5 promoted the reaction activity. The NiW/Al-KT-40 catalyst exhibited the highest HDN efficiencies compared with other catalysts, which is ascribed to the synergetic effect of the high specific surface area and suitable pore diameter, open ordered pore channel, high sulfidation, more acid sites, and moderate stacking degree of $WS_2$ phases [24,27,46].

**Table 5.** The hydrodenitrogenation conversion (HDNC) of quinoline on the NiW/Al-KT-X (X = 200, 80, 40, 10) and the NiW/KIT-5 catalysts.

| Catalyst | HDNc (%) | | | k ($h^{-1}$) | | | $r_{HDN}$ ($1 \times 10^{-4}$ mol $h^{-1}g^{-1}$) | | |
|---|---|---|---|---|---|---|---|---|---|
| | 340 °C | 360 °C | 380 °C | 340 °C | 360 °C | 380 °C | 340 °C | 360 °C | 380 °C |
| NiW/Al-KT-10 | 85.81 | 87.77 | 89.87 | 3.67 | 3.95 | 4.30 | 1.62 | 1.75 | 1.90 |
| NiW/Al-KT-40 | 90.83 | 91.49 | 95.14 | 4.49 | 4.63 | 5.69 | 1.99 | 2.05 | 2.52 |
| NiW/Al-KT-80 | 87.48 | 89.07 | 90.57 | 3.91 | 4.16 | 4.44 | 1.73 | 1.84 | 1.96 |
| NiW/Al-KT-200 | 81.52 | 85.36 | 90.00 | 3.17 | 3.61 | 4.33 | 1.41 | 1.60 | 1.92 |
| NiW/KIT-5 | 75.68 | 82.44 | 87.51 | 2.66 | 3.27 | 3.91 | 1.18 | 1.45 | 1.73 |

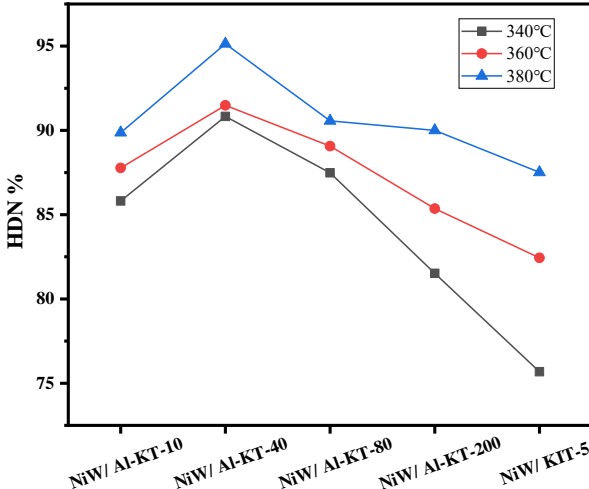

**Figure 11.** Hydrodenitrogenation (HDN) efficiencies of various catalysts at different temperatures.

$HDN_C$ for the catalysts with Al-modified supports was higher than 81%, in particular for the NiW/Al-KT-40 catalyst, the $HDN_C$ was higher than 90%, which showed better performance than that reported about catalysts with Al-modified MCM-41 [24]. The highest $HDN_C$ was 85% for the catalyst with Al-modified MCM-41.

To further analyze the mechanism of the HDN reaction, the selectivity of the hydrogenation products at different temperatures was calculated. Since the mechanism of the HDN reaction is similar for all catalysts, the NiW/Al-KT-80 catalyst was randomly chosen as an example, and the results are presented in Table 6. Clearly, the selectivity of PB was much lower than PCH at different temperatures, indicating that pathway(II) was the dominating reaction path. The results clearly display that, with the increase of temperature, the selectivity of all the non-nitrogen-containing products (PCHE, PB, and PCH) increased gradually with the increase of temperature, whereas the selectivities of all the nitrogen-containing products (THQ1, THQ5, DHQ, and OPA) declined significantly, indicating that increasing temperature can not only increase the ring-opening activity but also promote the C-N bond breaking activity [4,8,29,57].

**Table 6.** Product distribution of the NiW/Al-KT-80 catalyst in quinoline HDN.

| Selectivity (%) | PCH | PB | PCHE | DHQ | THQ5 | Q | OPA | THQ1 |
|---|---|---|---|---|---|---|---|---|
| 340 °C | 57.28 | 20.97 | 2.85 | 1.47 | 2.31 | 4.95 | 2.19 | 7.96 |
| 360 °C | 58.58 | 24.01 | 3.86 | 1.04 | 2.07 | 3.70 | 0.97 | 5.77 |
| 380 °C | 59.99 | 25.24 | 6.03 | 0.86 | 1.63 | 3.02 | 0.72 | 2.52 |

## 3. Discussion

The results of the hydrodenitrification of quinoline demonstrated that the introduction of aluminum atoms into KIT-5 had a positive influence on the HDN activity of catalysts. According to a series of characterization results mentioned above, the HDN activities of catalysts had a close relation with the structural characteristics of the supports, acidities of the catalysts, sulfidation degree of active species, and the structure of active phases [37,57,58].

The pore structural properties greatly affected the dispersion of active metals and the molecular diffusion behavior in a channel [21]. The modified Al-KT-X supports still maintained the relatively orderly mesoporous channels. The specific surface area, pore volume, and pore diameter increased when aluminum atoms were introduced into the KIT-5 framework, leading to more active sites of the catalysts and higher catalytic activity. All Al-modified catalysts exhibited a higher HDN activity than the pure NiW/KIT-5 catalyst. This was due to the large pore size, pore volume, and highly specific

surface area of Al-modified catalysts, which promoted the transfer of reactants and products and reduced the diffusion resistance remarkably [4].

Similarly, the acid amount of the catalysts is of great importance for catalytic activity and product selectivity [32,59,60]. The acid amount of various catalysts first increased and then decreased with increasing aluminum. The NiW/Al-KT-40 catalyst possessed the highest acid amount and the highest hydrodenitrogenation activity, indicating that the acid sites were favorable for the improvement of the HDN activity of the catalyst.

In addition, the structure of the active phase catalysts had a significant impact on the catalytic activity, which is closely connected with the dispersion of $WS_2$ particles and the interaction between metal species and supports [27,59,60]. With the increase of the aluminum content in the catalysts, the sulfidation degree increased first and then decreased, and this indicates that an appropriate aluminum content can promote the interaction between the active metal and support leading to a deeper sulfidation for the catalyst, which improves the HDN catalytic activity.

## 4. Materials and Methods

### 4.1. Materials

We used hydrochloric acid (HCl; Kemio, Tianjin, China; 36–38%), sodium aluminate ($NaAlO_2$; Kemio, Tianjin, China; 98%), tetraethylorthosilicate (TEOS; Aldrich, Shanghai, China; 98%), poly(ethylene glycol)-block-poly(propylene glycol)-block-poly(ethylene glycol) (F127; MW = 12,600; $EO_{106}EO_{70}EO_{106}$; Aldrich, Shanghai, China), nickel nitrate hexahydrate (Tianjin Shentai chemical factory 98%), ammonium paratungstate (Kemio; Tianjin, China; 98%), carbon disulfide ($CS_2$; Merck, Beijing, China; 99%), quinoline (Kemio, Tianjin, China; 99%), cyclohexane (Merck, Beijing, China; 99%), and decalin (Kemio, Tianjin, China; 98%).

### 4.2. Catalyst Synthesis

The pure KIT-5 support was prepared following the procedure reported in the literature [35]. Typically, 5.0 g of F127 was dissolved in 250 mL of 0.5 M hydrochloric acid with vigorous stirring for 24 h at 45 °C; next, 25.0 g of TEOS was added into the mixture drop by drop and stirred constantly at 45 °C for 24 h. Afterward, the reaction solution was transferred into a Teflon-lined stainless steel autoclave and heated at 100 °C for 24 h. After the hydrothermal reaction was completed, the solution was cooled down to room temperature. The obtained sample was collected by filtering, drying in air at 100 °C for 12 h and calcining at 550 °C (1 °C min$^{-1}$) for 6 h for removal of the template.

The Al-KT-X (X represents Si/Al molar ratios of 10, 40, 80, and 200, respectively) samples were synthesized with aluminum isopropoxide as aluminum source. First, 25.0 g TEOS and 5.0 g F127 were added to 250 mL of 0.5 M HCL and stirred continuously to form the white suspension. Next, different amounts of the aluminum source were added into the above solution under stirring for 12 h at the same temperature. Subsequently, the solution was transferred into the autoclave for 24 h at 100 °C. Finally, the as-synthesized materials were obtained by filtering, drying, and calcining.

The corresponding NiW/Al-KT-X catalysts and NiW/KIT-5 were prepared using the one-step incipient impregnation method. Nickel nitrate hexahydrate and ammonium paratungstate were used as Ni and W sources for the catalysts, respectively. The loading of 35.0 wt% $WO_3$ and 4.2 wt% NiO were impregnated on the support overnight. After impregnation, each sample was dried at 100 °C for 12 h in an oven and calcined for 6 h at 550 °C (2 °C min$^{-1}$) in the muffle furnace. The catalysts obtained were denoted as NiW/Al-KT-10, NiW/Al-KT-40, NiW/Al-KT-80, NiW/Al-KT-200, and NiW/KIT-5.

Prior to the catalytic reaction, the catalyst was activated by sulphidation in situ with the 3 wt% $CS_2$ solution (cyclohexane as the solvent) at a flow rate of 2.4 mL h$^{-1}$ under a pressure of 3.8 MPa and a temperature of 150 °C for 4 h. Then, the temperature was increased to 350 °C with a heating rate of 1 °C min$^{-1}$ and held for 12 h at 350 °C. The sulfide catalysts were characterized to study their physical and chemical properties.

### 4.3. X-ray Diffraction

Powder X-ray diffraction (XRD) patterns of the samples were analyzed on a Rigaku MiniFlexII X-ray diffractometer Buker with Cu K$\alpha$ ($\lambda$ = 0.1541 nm) radiation with a step of 0.002° s$^{-1}$ over a range of 0.5–3.0° (2$\theta$) for the supports and 2° s$^{-1}$ over a range of 5–90° (2$\theta$) for the oxide precursors.

### 4.4. N$_2$ Physisorption

With the Tristar-3020 Micrometrics volumetric apparatus, the nitrogen physisorption isotherms were tested. The total specific surface area was calculated by the standard Brunauer–Emmett–Teller (BET) method. The pore distribution derived from the absorption branch were obtained by the Barett–Joyner–Halenda (BJH) method, and the pore volume was acquired by pore size distribution curves.

### 4.5. Fourier Transform Infrared (FTIR) and Py-IR

Fourier transform infrared (FTIR) spectra of the supports were analyzed at a resolution of 2 cm$^{-1}$ in the range of 400–4000 cm$^{-1}$ using the Thermos Fisher Scientific Nicolet-380 instrument. Brønsted and Lewis acid distribution of the oxide precursors catalysts were performed by an FTIR spectrometer using pyridine as a probe molecule. The Py-IR investigation was conducted at different desorbed temperatures (100, 200, and 300 °C), The pyridine adsorption infrared (Py-IR) spectra were recorded at 100, 200, and 300 °C reflecting weak acid, medium acid, and strong acid, respectively [28].

### 4.6. Scanning Electron Microscopy

The morphologies of all the samples were studied using scanning electron microscopy (SEM) on a JSM-7900 F apparatus. Additionally, the surface element contents of the materials were measured by the SEM equipped with energy dispersive spectroscopy (EDS).

### 4.7. High Resolution Transmission Electron Microscopy

The WS$_2$ morphologies over the sulfide catalysts were observed through the high resolution transmission electron microscopy (HRTEM) on a JEOL (JEM-2100F, Tokyo, Japan). The corresponding samples were evenly dispersed in the ethanol solution and dropped on the ultra-thin carbon supporting film. Subsequently, the samples were naturally dried.

We used the following formulas to calculate the average slab number of stacks ($\overline{N}$) and average length ($\overline{L}$) of WS$_2$ stacking layers [5,8,11]:

$$\overline{N} = \frac{\sum\limits_{i=1...t} n_i N_i}{\sum\limits_{i=1...t} n_i} \tag{1}$$

$$\overline{L} = \frac{\sum\limits_{i=1...t} n_i L_i}{\sum\limits_{i=1...t} n_i} \tag{2}$$

where $n_i$, $N_i$, and $L_i$ represent the number of slabs, stacking layers of a WS$_2$ unit and the slab length of the WS$_2$ unit, respectively

### 4.8. X-ray Photoelectron Spectroscopy

Prior to the X-ray photoelectron spectroscopy (XPS) analysis, all the sulfide catalysts were ground into powder in an Ar-filled glovebox and stored in a sealed bag to avoid reoxidation. Then, the sample was transferred to the chamber of the XPS instrument without exposure to air. The XPS spectra were obtained with a Thermo Escalab 250Xi spectrometer equipped with an Al K$\alpha$ source (1486.6 eV), operating at 15.0 kV and 8.6 mA. The operating pressure inside the analysis chamber was below 1.0 × 10$^{-7}$ Pa. Using the Al 2p band at 76.4 ev as a standard, the peak shift was corrected [5,61].

### 4.9. Catalytic Activity Evaluation

The HDN activities of the series NiW/Al-KT-X and NiW/KIT-5 catalysts were evaluated in a fixed-bed reactor with a feed of quinolone (Q) in decalin (0.5 wt% N). One gram of fresh catalyst of 20–40 mesh size was loaded into the stainless steel reaction tube. After sulphidation, the hydrodenitrogenation reaction was carried out under the pressure of 3.8 MPa with a $H_2$/oil of 1250 mL/mL, and a constant weight hourly space velocity (WHSV) of 3 $h^{-1}$. The reaction temperatures were 340–380 °C. All reaction products were collected by the condensing system at 12–24 h reaction time and analyzed. The nitrogen compounds in the products were qualitatively analyzed using a Agilent-7890A GC–MS equipped with a capillary column (HP-5MS). The nitrogen contents in the samples were quantitatively analyzed using a GC equipped with a flame ionization detector (FID) and a HP-5 column using an internal standard method. All the experiments were preformed two or three times with good repeatability.

The hydrogenation reaction network of quinoline has two pathways in Figure 12: the pathway(I) is Q → 1,2,3,4-tetrahydroquinoline (THQ1) → ortho-propylaniline (OPA) → propylbenzene (PB), and the other pathway(II) is Q → decahydroquinoline (DHQ) → 2-propyl-cyclohexylamine (PCHA) → propyl-cyclohexene (PCHE) + propyl-cyclohexane (PCH). Generally, the hydrodenitrogenation conversion ($HDN_C$) of each catalyst was calculated using the following equation [53–55]:

$$HDN_C(\%) = \frac{n_{PB} + n_{PCH} + n_{PCHE}}{n_Q + \sum n_i} \tag{3}$$

in which the Q, PB, PCH, and PCHE concentrations collected in the products are defined as $n_Q$, $n_{PB}$, $n_{PCH}$, and $n_{PCHE}$, respectively. $\sum n_i$ represents the sum of all product concentrations obtained from quinoline, including PB, THQ5, OPA, PCH, DHQ, THQ1, PCHE, and PCHA.

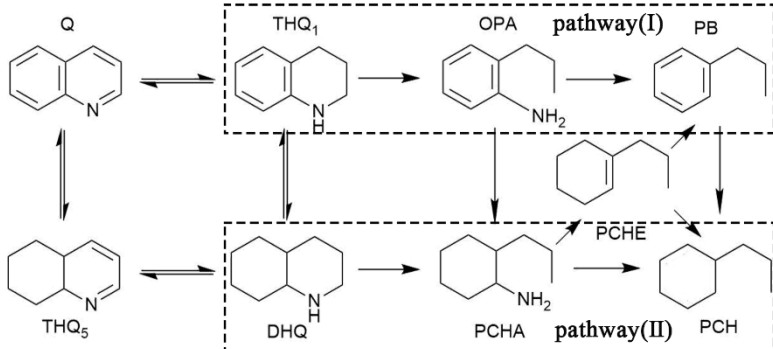

**Figure 12.** The HDN reaction network of quinoline.

The reaction rates ($r_{HDN}$) and the rate constants (k) of the series of catalysts for quinoline denitrogenation were calculated by the following formulas based on the pseudo first-order kinetics [61–63]:

$$r_{HDN} = -\frac{F_{A0}}{v_R}(1 - HDN_C)\ln(1 - HDN_C) \tag{4}$$

$$k = -\frac{F_{A0}}{C_{A0}v_R}\ln(1 - HDN_C) \tag{5}$$

where $r_{HDN}$ represents the reaction rate (mol $s^{-1}$ $g^{-1}$), k represents the rate constant ($s^{-1}$), $C_A$ and $C_{A0}$ are the concentration of quinoline (mol $L^{-1}$) in the product and feed, $V_R$ represents the volume of catalysts (L), and $F_{A0}$ stands for the molar flow rate of quinoline (mol $s^{-1}$).

## 5. Conclusions

A series of Al-KT-X and KIT-5 materials were successfully synthesized using the direct hydrothermal method, and they supported the NiW active phases for the quinoline hydrodenitrification reaction. All supports and catalysts were characterized by XRD, $N_2$ isotherm absorption–desorption, FTIR, Py-IR, and SEM, and the series of sulfide catalysts were characterized by HRTEM and XPS. The results showed that the addition of aluminum atoms did not destroy the orderly mesoporous structure of KIT-5 and exhibited a larger pore size, pore volume, and specific surface area. Among all the Al-modified supports, the Al-KT-40 support possessed a suitable surface area (1057.15 $m^2g^{-1}$), pore volume (0.64 $cm^3 \cdot g^{-1}$), and orderly 3D channel (5.43 nm). The addition of aluminum atoms into the framework of KIT-5 resulted in an increase of acid sites and a good distribution of active phases. Overall, the modified NiW/Al-KT-X catalysts were superior in activity compared with the pure NiW/KIT-5 catalyst, which was attributed to the large pores, more acidic sites, and more sulfided active metals. The NiW/Al-KT-40 catalysts exhibited the highest HDNc of 95.14% at 380 °C.

**Author Contributions:** X.L. (Xing Liu) designed and performed the experiments; X.L. (Xing Liu) and H.C. analyzed the data; Z.L., X.L. (Xin Li), L.Y., and H.D. provided technical support; X.L. (Xing Liu) wrote the paper; S.G. reviewed and edited the paper; and L.Z. provided financial support. All authors have read and agreed to the published version of the manuscript.

**Funding:** This research was funded by the Science and Technology Major Project of Shanxi Province (No. 20181101018), Bidding Project of Shanxi Province (No. 20191101001) and the Shanxi Science and Technology Department (No. 202001101012).

**Conflicts of Interest:** The authors declare no conflict of interest.

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
