# Peer review of "Synthesis of NiW Supported on an Al-Modified Cubic Ia3d Mesoporous KIT-5 Catalyst and Its Hydrodenitrogenation Performance of Quinoline"

_catalysts, doi:10.3390/catal10101183_

Round 1
Reviewer 1 Report
The proposed manuscript represents interesting results but some of them need major revision:
- The information included in the abstract is misleading. The abstract does not provide any overview of the presented results. This part must be improved.
- The introduction part must be extended. More literature review should be conducted and summarized, e.g. alumina catalyst, modification with fluorine, phosphorus, molybdenum etc.
- In my opinion, the manuscript should be clearly divided into two parts: an investigation of supports and final catalysts.
- Presentation of data must be improved: SEM and TEM images suffer from invisible scale, in my opinion, SEM images do not provide any information of differences in features of prepared supports or aluminium distribution, TEM images should be further processed in terms of determining of interplanar spacing and thus evaluating of Al effect on the structure.
- It is not clear why reflections attributed to nickel and tungsten phases are not visible at the wide-angle XRD? The discussion should be extended. In my opinion, the content of 35 wt % of WO3 cannot be misled.
- The final content of Al, Ni and W must be determined.
- More details regarding the experimental section should be provided. How long did the catalytic test last? Was it necessary to use the condensing system for liquid products? What about the repeatability of experiments?
- What about the stability of the catalysts?
- The conclusion section must be extended, results must be quantified and compared with the literature results.
- The whole manuscript should be revised in terms of English and used expressions. Phrases such as the amount of acid (line 17), the existence of highly crystalline (line 141) are not acceptable. Some typos or double spaces are also present.
Author Response
Comment 1: The information included in the abstract is misleading. The abstract does not provide any overview of the presented results. This part must be improved.
Response 1: We thanks the reviewer for the suggestions and have improved the abstract in the revised manuscript.
Comment 2: The introduction part must be extended. More literature review should be conducted and summarized, e.g. alumina catalyst, modification with fluorine, phosphorus, molybdenum etc.
Response 2: Thanks very much for your valuable suggestion. The introduction has been further elaborated and more literature have been cited in the introduction part. The relative description has been added in the introduction part in the revised manuscript.
Comment 3: In my opinion, the manuscript should be clearly divided into two parts: an investigation of supports and final catalysts.
Response 3: Thanks very much for your constructive suggestion. The section of results has been revised in revised manuscript and divided into two parts including the section of 2.1 (Characteristics of the supports) and the section of 2.2(Characteristics of the catalysts) according to your comment.
Comment 4: Presentation of data must be improved: SEM and TEM images suffer from invisible scale, in my opinion, SEM images do not provide any information of differences in features of prepared supports or aluminum distribution, TEM images should be further processed in terms of determining of interplanar spacing and thus evaluating of Al effect on the structure.
Response 4: Thanks very much for your kind reminding. We are really sorry for the unclear scales in SEM and TEM images and they have been improved in the revised manuscript.
As for the SEM images, the differences in features of prepared support can be observed in Figure 4. The morphologies of all the supports are obviously irregular. Moreover, the surface of the pure KIT-5 is smoother than that of the Al-modified materials. As for the EDS image, the EDS pattern of all the supports show a uniformly distribution of Al in the surface of supports with no obvious difference. Therefore, the SEM-EDS image of Al-KT-40 sample was randomly selected in this paper.
As for the TEM images, it has been further processed and the interplanar spacing in TEM images with different supports is calculated by the Nano Measurer. The result shows that the interplanar spacing of supports is almost similar, indicating that the addition of aluminum atoms has no obvious effect on the structure of supports. These results agree well with N2 adsorption and XRD results. The relative description has been added in the revised manuscript.
Comment 5: It is not clear why reflections attributed to nickel and tungsten phases are not visible at the wide-angle XRD? The discussion should be extended. In my opinion, the content of 35 wt % of WO3 cannot be misled.
Response 5: Thank you for the suggestions. It is true that the content of 35 wt % of WO3 is so high that the tungsten phases should be visible at the wide-angle XRD. However, the wide-angle XRD results show that the diffraction pattern of Al-modified NiW/KIT-5 catalysts has no apparent characteristic diffraction peaks compared with the pure catalyst, indicating that active species have better dispersion and smaller crystallites. Meanwhile, similar results have been reported in previous studies (catalysis letters, 144(2014) 1584–1593; RSC advance, 6 (2016) 61747–61757). The relative description has been added in revised manuscript.
Comment 6: The final content of Al, Ni and W must be determined.
Response 6: Thanks very much for your valuable advice. Indeed, it had better to determine the final content of Al, Ni and W of the catalysts. However, since impregnation method is the most common to prepare catalyst and all active metal could basically be loaded in the support, the final content of metal element is not determined. So, we did not show them in this manuscript. We are sorry for this.
Comment 7: More details regarding the experimental section should be provided. How long did the catalytic test last? Was it necessary to use the condensing system for liquid products? What about the repeatability of experiments?
Response 7: Thanks very much for your kind reminding. More details regarding the experimental section have been provided and marked in red color in the revised manuscript. Also, it is necessary to use the condensing system for collecting the liquid products during the reaction. All the experiments have been done 2 or 3 times with good repeatability. The relative description has been added in the experimental section in the revised manuscript.
Comment 8: What about the stability of the catalysts?
Response 8: Thank you for your valuable advice. Actually, the stability of the catalysts is important and the information about the stability of the catalysts should be provided. However, more attention has been paid on the preparation of the support and the catalysts in this paper. And the reaction activity as well as the stability of the catalysts will be presented in another paper in detail. Thanks for your understanding.
Comment 9: The conclusion section must be extended, results must be quantified and compared with the literature results.
Response 9: Thanks very much for your advice. The conclusion section has been extended and the results have been quantified and compared with the literature results according to your suggestion.
Comment 10: The whole manuscript should be revised in terms of English and used expressions. Phrases such as the amount of acid (line 17), the existence of highly crystalline (line 141) are not acceptable. Some typos or double spaces are also present.
Response 10: Thank you for your valuable advice. The whole manuscript has been revised in terms of English and used expressions. The phrases of "the amount of acid" and "the existence of highly crystalline" have been changed to "acidity" and " the formation of of larger crystallites". Some mistakes such as typos or double spaces have been corrected in the revised manuscript. We are really sorry for our improper writing. We have rechecked the whole manuscript to avoid the similar mistakes.
Reviewer 2 Report
In this paper entitled “Synthesis of NiW supported on Al-modified cubic Ia3d mesoporous KIT-5 catalyst and its hydrodenitrogenation performance of quinoline”, Liangfu Zhao and collaborators reported the preparation of NiW catalysts supported on pure KIT-5 and series of Al-modified KIT-5 materials, named Al-KT-X, having different silicon-aluminum ratio (X = 0, 40, 80 and 200). All the obtained catalysts were characterized by XRD, N2 isotherm absorption-desorption, FT-IR, Py-IR, SEM, HRTEM and XPS analyses, and then tested for the reaction of quinoline hydrodenitrification (HDN); all the NiW/Al-KT-X catalysts exhibited better performances than NiW/KIT-5, and in particular NiW/Al-KT-40 system showed maximal HDN conversion. I found this work very interesting: the paper is well written and the results are very clearly presented and discussed. Therefore, I believe that the manuscript is worth to be published in the MDPI journal Catalysts after few minor revisions.
1. Concerning the investigation of quinoline HDN reaction, catalytic activity was found to increase with temperature. However, authors performed tests only at three different temperatures: 340, 360 and 380 °C, and for all catalysts the best performances were found at 380 °C. Why did the authors not used also higher reaction temperatures?
2. I suggest to add a couple of relevant references in the introduction: a) concerning the excellent mechanical performances, low price and higher thermal stability of γ-alumina, DOI: 10.1002/slct.201601736; b) concerning other mesoporous materials (in addition to those already mentioned in the paper) as a research hotspot of the catalyst supports, DOI: 10.1016/j.molcata.2011.09.002 and DOI: 10.1039/C6GC02367C.
3. Concerning figures, I suggest to enlarge the font size of letters and numbers in both the axes and graphs, since they are often so small to be hard to read.
Author Response
Comment 1: Concerning the investigation of quinoline HDN reaction, catalytic activity was found to increase with temperature. However, authors performed tests only at three different temperatures: 340, 360 and 380 °C, and for all catalysts the best performances were found at 380 °C. Why did the authors not used also higher reaction temperatures?
Response 1: Thank you for your suggestions. The result shows that the increase of temperature is beneficial to the enhancement of HDN catalytic activity, which agrees with many literatures about HDN reaction. In this paper, the temperature of 340, 360 and 380 °C have been selected to evaluate the catalysts, which can be compared with the literature results at the same temperature (Energy Fuels, 33 (2019) 1450−1457; Industrial & Engineering Chemistry Research 56 (2017) 10287−10299). In addition, because reaction temperature for HDN reaction in industry is generally lower than 380 ℃, the higher reaction temperature has not been used in this study.
Comment 2: I suggest to add a couple of relevant references in the introduction: a) concerning the excellent mechanical performances, low price and higher thermal stability of γ-alumina, DOI: 10.1002/slct.201601736; b) concerning other mesoporous materials (in addition to those already mentioned in the paper) as a research hotspot of the catalyst supports, DOI: 10.1016/j.molcata.2011.09.002 and DOI: 10.1039/C6GC02367C.
Response 2: Thanks very much for your valuable suggestion. The relative references have been added in the revised manuscript.
Comment 3: Concerning figures, I suggest to enlarge the font size of letters and numbers in both the axes and graphs, since they are often so small to be hard to read.
Response 3: Thank you for your reminding. We are sorry for the unclear font size of letters and numbers in both the axes and graphs and they have been corrected in the revised manuscript.
Round 2
Reviewer 1 Report
The manuscript is now acceptable. I have no substantive comments, but the manuscript still requires corrections in terms of photo quality and scale on microscopic images.